# MASSIVELY PARALLEL HYPERPARAMETER TUNING

## ABSTRACT

Modern machine learning models are characterized by large hyperparameter search spaces and prohibitively expensive training costs. For such models, we cannot afford to train candidate models sequentially and wait months before finding a suitable hyperparameter configuration. Hence, we introduce the *large-scale regime* for parallel hyperparameter tuning, where we need to evaluate orders of magnitude more configurations than available parallel workers in a small multiple of the wall-clock time needed to train a single model. We propose a novel hyperparameter tuning algorithm for this setting that exploits both parallelism and aggressive early-stopping techniques, building on the insights of the Hyperband algorithm (Li et al., 2016). Finally, we conduct a thorough empirical study of our algorithm on several benchmarks, including large-scale experiments with up to 500 workers. Our results show that our proposed algorithm finds good hyperparameter settings nearly an order of magnitude faster than random search.

## 1 INTRODUCTION

Although machine learning models have recently achieved dramatic successes in a variety of practical applications, these models are highly sensitive to internal parameters, i.e., *hyperparameters*. In this modern era of machine learning, three trends motivate a new approach to hyperparameter tuning:

1. **High-dimensional hyperparameter spaces**. Machine learning models are becoming increasingly complex, as evidenced by modern neural networks with dozens of hyperparameters, e.g., number of layers, width of each layer, forms of regularization, types of nodes, type of activation nonlinearities, etc. For such complex models with hyperparameters that interact in unknown ways, a practitioner is forced to evaluate potentially thousands of different hyperparameter settings.

2. **Increasing training times**. As datasets grow larger and models become more complex, training a model has become dramatically more expensive, often taking days or weeks on specialized high-performance hardware. For instance, Google's neural machine translation system takes one week to train using eight GPUs in parallel (Wu et al., 2016) and Google's LSTM acoustic model (Sak et al., 2014) also takes a few weeks to train. This trend is particularly onerous in the context of hyperparameter tuning, as a new model must be trained to evaluate each putative hyperparameter configuration.

3. **Rise of parallel computing**. The combination of a growing number of hyperparameters and longer training time per model preclude the use of sequential hyperparameter tuning methods; we simply cannot wait months or years to find a suitable hyperparameter setting. Thankfully, the expansion of cloud computing resources has made specialized hardware like GPUs and TPUs (Dean and Hölzle, 2017) widely accessible. Leveraging these parallel and distributed computational resources presents an attractive path forward to combat the increasingly challenging problem of hyperparameter optimization.

Our goal is to design a hyperparameter tuning algorithm that can effectively leverage parallel resources. This goal seems trivial with standard methods like random search, where we could train different configurations in an embarrasingly parallel fashion. However, in practice, the number of putative configurations often dwarfs the number of available parallel resources. Hence, we aim to tackle the following problem, which we refer to as the **large-scale regime** for hyperparameter optimization:

*Evaluate orders of magnitude more hyperparameter configurations than available parallel workers in a small multiple of the wall-clock time needed to train a single model.*

Existing parallel hyperparameter tuning methods generally fall into two categories: (1) embarrassingly parallel methods that use a fixed sampling scheme (i.e. random search and grid search); and (2) adaptive configuration selection schemes using Bayesian optimization. Methods in the first category are ill suited for the large-scale regime because they can only evaluate configurations on the order of number of parallel workers. Methods in the second category are general black-box optimization routines that are complex and computationally expensive. In contrast, our proposed method empirically exhibits state-of-the-art performance without these added layers of complexity.

Our main contribution in this work is a practical hyperparameter tuning algorithm for the large-scale regime that exploits parallelism and aggressive early-stopping techniques. We build upon the *Hyperband* algorithm (Li et al., 2016), adapting it for the parallel setting and improving it in two significant ways, as described in Section 3.1. Furthermore, we conduct a thorough empirical study of our algorithm on multiple benchmarks, demonstrating success in the large-scale regime on tasks with up to 500 workers. Lastly, we present a small case study on the efficient use of parallel computing resources for hyperparameter tuning.

## 2 RELATED WORK

We review prior work on hyperparameter tuning in both the sequential and parallel settings. While there are many sequential algorithms that have associated implementations for the parallel setting, we categorize algorithms not specifically designed for the parallel setting as sequential methods.

### 2.1 SEQUENTIAL METHODS

Existing hyperparameter tuning methods attempt to speed up the search for a good configuration by either adaptively selecting configurations or adaptively evaluating configurations. Adaptive configuration selection approaches attempt to identify promising regions of the hyperparameter search space from which to sample new configurations to evaluate. Well-established configuration selection approaches include SMAC, Spearmint, TPE, and GP-UCB (Hutter et al., 2011; Snoek et al., 2012; Bergstra et al., 2011; Srinivas et al., 2010), which have been shown to outperform random search in empirical studies. However, by relying on previous observations to inform which configuration to evaluate next, these algorithms are inherently sequential and thus not suitable for the large-scale regime, where the number of updates to the posterior is limited.

Adaptive configuration evaluation approaches attempt to early-stop poor configurations and allocate more training "resources" to promising configurations. The use of resources can, for example, be measured as the of number of training iterations, cpu or wall-clock time, or the number of training examples. Previous methods like György and Kocsis (2011); Agarwal et al. (2011); Sabharwal et al. (2016) provide theoretical guarantees under strong assumptions on the convergence behavior of intermediate losses. Krueger et al. (2015) relies on a heuristic early-stopping rule based on sequential analysis to terminate poor configurations.

Hybrid Bayesian optimization approaches that combine adaptive configuration selection and evaluation include Swersky et al. (2013; 2014); Domhan et al. (2015); Klein et al. (2017a). However, these methods scale poorly with number of configurations due to the cubic complexity of fitting a Gaussian process (Snoek et al., 2015). Nonetheless, Klein et al. (2017a) recently introduced Fabolas and demonstrated state-of-the-art performance on several hyperparameter tuning tasks; we will provide a detailed comparison to Fabolas in Section 4.

Lázaro-Gredilla et al. (2010); Wilson et al. (2015) improve the scaling profiles of Gaussian processes by imposing additional structure on the model. Additionally, Bayesian deep neural networks, which scale linearly as a function of the number of observations, have been used to model the posterior (Snoek et al., 2015; Springenberg et al., 2016). However, the improved computational tractibility comes at a price; for a fixed number of evaluations, these methods generally underperform vanilla Gaussian processes from an accuracy perspective.

Li et al. (2016) proposed Hyperband, an adaptive configuration evaluation approach which does not have the aforementioned drawbacks and showed that it achieves state-of-the-art performance on several empirical tasks. At first glance, the Hyperband algorithm appears trivial to parallelize since all evaluated configurations are drawn independently. However, the algorithm eliminates

configurations at an exponential rate, severely limiting the degree of parallelism in later elimination rounds. Additionally, these eliminations are performed at fixed intervals, making the algorithm susceptible to stragglers. Moreover, the divide-and-conquer strategy of running an instance of the algorithm on each machine does not reduce the latency of receiving an output from the algorithm. Our proposed algorithm adapts Hyperband for the large-scale regime by addressing these issues.

Note that Hyperband uses randomly sampled hyperparameter configurations by default, but can be combined with any sampling mechanism. For example, Klein et al. (2017b) combined Bayesian neural networks with Hyperband by first training a Bayesian neural network to predict learning curves and then using the model to select promising configurations to use as inputs to Hyperband.

## 2.2 PARALLEL METHODS

Grid search and random search are popular in practice because they are trivial to parallelize. Nonetheless, distributed versions of SMAC, Spearmint, and TPE (Hutter et al., 2011; Snoek et al., 2012; Bergstra et al., 2011) are sometimes used. These methods either act greedily (Spearmint) or rely on randomness to provide additional configurations to evaluate (SMAC, TPE). More sophisticated batch Bayesian optimization methods attempt to construct a batch of configurations by optimizing a joint objective (Shah and Ghahramani, 2015; González et al., 2016; Wu and Frazier, 2016). However, these methods fair poorly in the large-scale regime since all configurations are trained to completion.

As for hybrid adaptive configuration selection and evaluation methods, the hybrid sequential methods mentioned in Section 2.1 can be parallelized using constant liar (CL) type heuristics (Ginsbourger et al., 2010; González et al., 2016). These heuristics fill in the objective value of pending configurations with a constant before proposing new configurations to try. Note that assuming the Bayesian model is correct, algorithms parallelized using the CL heuristic are inferior to the sequential version, which uses a more accurate posterior, updated with true objective values, to propose new points.

Similar to the sequential method in Domhan et al. (2015), Rasley et al. (2017) uses parametric learning curves to perform early-stopping but adds a scheduling algorithm to effciently use parallel resources. More recently Golovin et al. (2017) introduced Vizier, a black-box optimization service with support for multiple tuning methods and early-stopping options. For succinctness, we will refer to Vizier's default Bayesian optimization algorithm as "Vizier" with the understanding that it is simply one of methods available on the Vizier platform. We compare to Vizier in Section 4.3.

Another class of related methods focus on learning neural network architectures for a given task in a large-scale setting (Zoph and Le, 2017; Real et al., 2017). Real et al. (2017) uses evolutionary algorithms to design neural network architectures for image classifiers. A different approach is taken in Zoph and Le (2017), which uses reinforcement learning to learn a policy used to generate neural network architectures that perform well on a given task. While these method match or even outperform human designed architectures, they are computationally expensive and require task dependent specifications on how architectures can be modified to arrive at new architectures. We note that Zoph and Le (2017) achieves state-of-the-art performance on the Penn Treebank language modeling task, outperforming previous methods by a large margin; we consider a similar tuning task in Section 4.3. However, they explore a much larger search space and the proposed architecture still requires hyperparameter tuning of learning parameters, weight decay, and batchnorm epsilon.

## 3 ALGORITHM

Hyperband (Li et al., 2016) calls the Successive Halving algorithm (Karnin et al., 2013; Jamieson and Talwalkar, 2015), with different early-stopping rates as a subroutine. Li et al. (2016) showed that Successive Halving with aggressive early-stopping matches or outperforms Hyperband for a wide array of hyperparameter optimization tasks. Additionally, for modern day hyperparameter tuning problems with high-dimensional search spaces and models with high training cost, aggressive early-stopping is *necessary* for the problem to be tractable. Hence, we focus on adapting the synchronous Successive Halving algorithm (SHA) for the parallel setting. Note that parallelizing Hyperband is trivia after parallelizing SHA—simply take the best performing configuration across multiple brackets of SHA with different early-stopping rates.

---

**Algorithm 1:** Asynchronous Successive Halving Algorithm.

**Input:** $r$, $\eta$ (default $\eta = 3$), $s$
**Algorithm** `async_SHA()`

  **repeat**
    **for** free worker **do**
      $(\theta, k) = $ `get_job()`
      worker performs `run_then_return_val_loss`$(\theta, r\eta^{s+k})$
    **end**
    **for** completed job $(\theta, k)$ with loss $l$ **do**
      Update configuration $\theta$ in rung $k$ with loss $l$.
    **end**

  **Procedure** `get_job()`
    `// A configuration in a given rung is "promotable" if its`
    `validation loss places it in the top` $1/\eta$ `fraction of completed`
    `configurations in its rung and it has not already been promoted.`
    Let $\theta_k$ be the furthest trained *promotable* configuration $\theta$, with $k$ indicating its rung.
    **if** $\theta_k$ exists **then**
      Promote $\theta_k$ to rung $k + 1$.
      **return** $\theta_k, k + 1$
    **else**
      Add a new configuration $\theta_0$ to bottom rung.
      **return** $\theta_0, 0$
    **end**

---

In Algorithm 1, we introduce a "conservative" asynchronous algorithm, which has the property of advancing all configurations that would have been advanced by synchronous SHA. Thus, the final selected configuration is guaranteed to be at least as good as the one returned by the synchronous algorithm, at the expense of a potentially larger total computational cost.

Algorithm 1 requires as input a minimum resource $r$, the reduction factor $\eta > 1$, and the minimum early-stopping rate $s$. We will refer to trials of SHA with different values of $s$ as *brackets* and, within a bracket, we will refer to each round of promotion as a *rung* with the base rung numbered 0 and increasing. For a given $\eta$, only the top $1/\eta$ fraction of configurations are promoted to the next rung. For a given $s$, a minimum resource of $r\eta^s$ will be allocated to each configuration. Hence, lower $s$ corresponds to more aggressive early-stopping, with $s = 0$ prescribing a minimum resource of $r$. Another component of the algorithm is the `run_then_return_val_loss`$(\theta, r)$ subroutine, which returns the validation loss after training the model with the hyperparameter setting $\theta$ and resource allocation $r$. The subroutine is asynchronous and the code execution of `async_SHA` continues after the job is passed to the worker.

Asynchronous SHA promotes configurations to the next rung whenever possible instead of waiting for a rung to complete before proceeding to the next rung. Additionally, if no promotions are possible, the asynchronous algorithm simply adds a configuration to the base rung, so that more configurations can be promoted to the upper rungs. This promotion scheme is laid out in the `get_job` subroutine.

From the promotion scheme, it is clear that Algorithm 1 is asynchronous because a new job can be requested at any time and does not require waiting for other jobs to complete. Thus, no training job or advancement decision is blocked due to stragglers. Of course, this also means we may prematurely advance some configurations, but the probability of incorrect promotions decreases as more configurations are evaluated.

Note there is no bound on the number of rungs in a bracket and, therefore, no bound on the maximum resource that can be allocated to a given configuration. However, an upper bound on the maximum resource is often desired (i.e. limit the number of epochs of stochastic gradient descent when training a neural network to prevent overfitting (Hardt et al., 2016)). Algorithm 1 can be easily adapted to enforce a maximum resource per configuration by limiting the number of rungs. Specifically, for a given maximum resource $R$ and bracket $s$, limit the number of rungs to $\log_\eta(R/r) - s + 1$.

Finally, since Hyperband simply runs multiple brackets of SHA, we can parallelize Hyperband by taking the minimum across multiple brackets of asynchronous SHA. In practice, if prior knowledge is available about the task, a user can run a specific bracket with a target early-stopping rate instead of running all possible brackets as is done by Hyperband. In our experience, early-stopping is helpful for most neural network type tasks and brackets with minimum resources of $R/64$ or even $R/256$ are reasonable for a given maximum resource per configuration of $R$.

## 3.1 Algorithm Discussion

Asynchronous SHA improves upon the synchronous version of the algorithm in two important ways:

1. Li et al. (2016) has two versions of Successive Halving, one for the finite horizon (bounded resource per configuration) and another for the infinite horizon setting. Asynchronous SHA consolidates the two settings into one algorithm. For the finite horizon setting, simply limit the number of rungs per bracket. For the infinite horizon setting, the maximum resource per configuration increases naturally as configurations are promoted to higher rungs. In contrast, the synchronous version of the algorithm relies on the doubling trick and reruns brackets with larger budgets to increase the maximum resource.

2. The algorithm analyzed in Li et al. (2016) differs from the algorithm used in practice. The theoretical analysis relies on the budget per bracket increasing so that the number of evaluated configurations $n$ will eventually be large enough to find a sufficiently good configuration. In contrast, the algorithm recommended for use in practice runs a bracket with a fixed budget multiple times so that the waiting time for a configuration trained on the maximum resource does not double with each outer loop. Asynchronous SHA naturally grows the number of configurations and trains a configuration on the maximum resource whenever possible, so we do not need to rerun brackets or wait exponentially longer for configurations trained to completion.

A theoretical treatment of asynchronous SHA is out of the scope of our paper and reserved for future work. However, we conjecture that the theoretical guarantees for synchronous SHA will also apply for our proposed algorithm.

## 4 Empirical Evaluation

In this section, we first present experiments in the sequential setting to provide a point of comparison for asynchronous SHA and to compare Hyperband with a recent Bayesian optimization method combining adaptive configuration selection and evaluation method. Next, we show that asynchronous SHA achieves comparable performance as the serial version of the algorithm and is not hurt too much by incorrect promotions. Finally, we present large-scale experiments using asynchronous SHA on a language modeling task and a speech acoustic modeling task.

## 4.1 Comparison with Fabolas in Sequential Setting

Klein et al. (2017a) showed that Fabolas can be over an order of magnitude faster than existing Bayesian optimization methods. Additionally, the empirical studies presented in Klein et al. (2017a) suggest that Fabolas is faster than Hyperband at finding a good configuration. We conducted our own experiments to compare Fabolas with Hyperband on the following tasks: (1) tuning an SVM using the same search space as Klein et al. (2017a), (2) tuning a convolutional neural network (CNN) with the same search space as Li et al. (2017) on CIFAR-10 (Krizhevsky, 2009), and (3) tuning a CNN on SVHN (Netzer et al., 2011) with varying number of layers, batch size, and number of filters (see Appendix A.1 for more details). In the case of the SVM task, the allocated resource is number of training datapoints, while for the CNN tasks, the allocated resource is the number of training iterations. We note that Fabolas was specifically designed for data points as the resource, and hence, is not directly applicable to tasks (2) and (3). However, freeze-thaw Bayesian optimization (Swersky et al., 2014), which was specifically designed for models that use iterations as the resource, is known to perform poorly on deep learning tasks (Domhan et al., 2015). Hence, we believe Fabolas to be a reasonable competitor for tasks (2) and (3) as well, despite the aforementioned shortcoming.

We find that when using intermediate losses recorded by Hyperband to track the best performing configuration, Hyperband actually outperforms Fabolas. Additionally, as mentioned in Section 2.1,

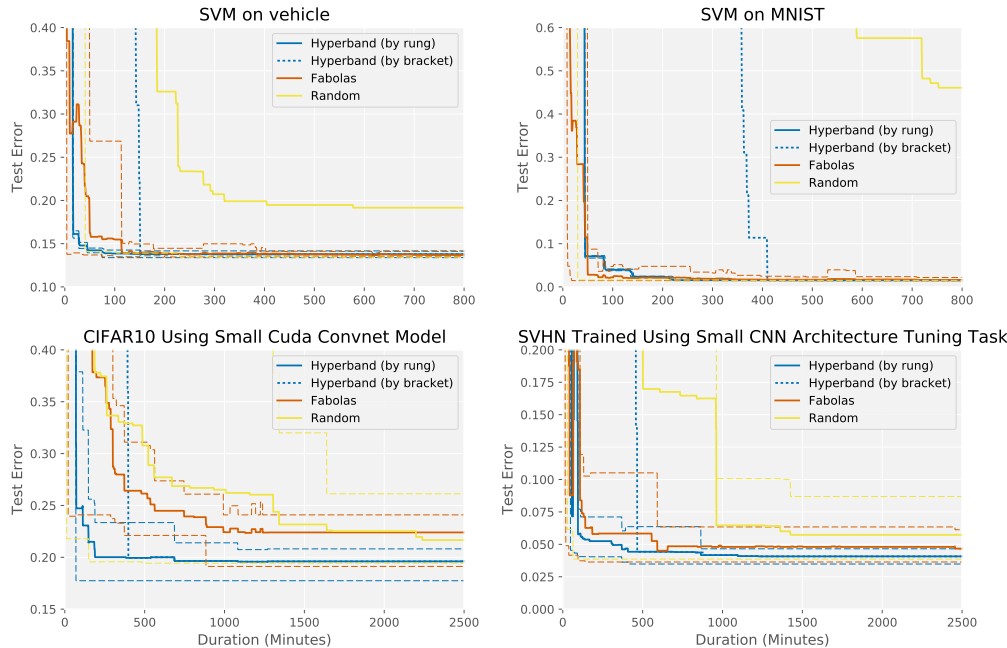

Figure 1: **Sequential Experiments** (1 worker) with Hyperband running synchronous SHA. Hyperband (by rung) records the incumbent after the completion of a SHA rung, while Hyperband (by bracket) records the incumbent after the completion of an entire SHA bracket. The average test error across 10 trials of each hyperparameter optimization method is shown in each plot. Dashed lines represent min and max ranges for each tuning method.

while Fabolas can be parallelized using the constant liar heuristic, the distributed version of the algorithm would at best achieve similar performance to the sequential algorithm. Hence, we will focus on evaluating asynchronous SHA in the next two sections.

For these sequential experiments, we use the same evaluation framework as Klein et al. (2017a), where the best configuration, also known as the *incumbent*, is recorded through time and the test error is calculated in an offline validation step. Following Klein et al. (2017a), the incumbent for Hyperband is taken to be the configuration with the lowest validation loss and the incumbent for Fabolas is the configuration with the lowest predicted validation loss on the full dataset. We use this framework to reproduce the results for Hyperband, Fabolas, and random search on the SVM task from Klein et al. (2017a) and apply the same methodology to the two CNN tuning tasks. We will refer to Hyperband as evaluated in Klein et al. (2017a) as "Hyperband (by bracket)," since the incumbent is recorded after the completion of each SHA bracket. We also evaluate the performance of Hyperband when recording the incumbent after the completion of each rung of SHA instead of each bracket to make use of intermediate validation losses; we will refer to Hyperband using this accounting scheme as "Hyperband (by rung)."

In Figure 1, we show the performance of Hyperband, Fabolas, and random search. Our results show that Hyperband (by rung) is competitive with Fabolas at finding a good configuration and will often find a better configuration than Fabolas with less variance. Note that Hyperband loops through the brackets of SHA, ordered by decreasing early-stopping rate. Hence, most of the progress made by Hyperband comes from the brackets with the most aggressive early-stopping rates. This supports our claim in the beginning of Section 3 and further validates our focus on asynchronous SHA in the remainder of our experiments.

## 4.2 LIMITED-SCALE DISTRIBUTED EXPERIMENTS

We evaluate asynchronous SHA on tasks (2) and (3) described in Section 4.1 to see how the asynchronous algorithm compares. For both tasks, we use asynchronous SHA with a bounded maximum resource (number of iterations) per configuration $R$ and $\eta = 4$. The minimum resource $r$ is set to

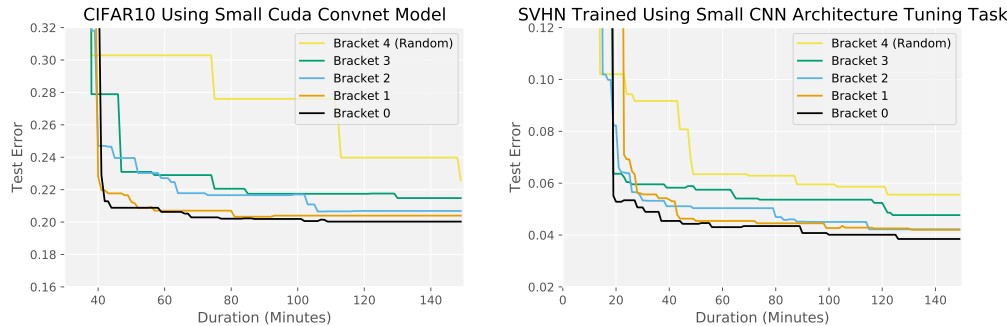

Figure 2: **Limited-scale distributed experiments** with 25 workers using asynchronous SHA. For each bracket, the average test error across 5 trials is shown in each plot.

$R\eta^{-4}$ for 5 possible brackets. For each experiment, asynchronous SHA is run with 25 workers for $3\times$ the time needed to train a single model.

Figure 2 shows the average test error of brackets of asynchronous SHA with different early-stopping rates. For the CIFAR-10 task, it took Hyperband over 200 minutes in the sequential setting to find a good configuration (Figure 1), while it took Bracket 0 slightly over 40 minutes in the parallel setting. For the SVHN task, it took Hyperband 300 minutes to find a configuration with a test error below $5\%$ (Figure 1), whereas Bracket 0 and Bracket 1 in the parallel setting found one in less than 40 minutes. On both tasks, asynchronous SHA succeeded in the large-scale regime; Bracket 0 evaluated over one thousand configurations in 40 minutes with just 25 workers and found a good configuration in approximately the time needed to train a single model. We note that the speedup is less than the number of machines due to the moderate difficulty of the problem. We explore harder tasks with a much higher degree of parallelism in the next section.

## 4.3 LARGE-SCALE EXPERIMENTS

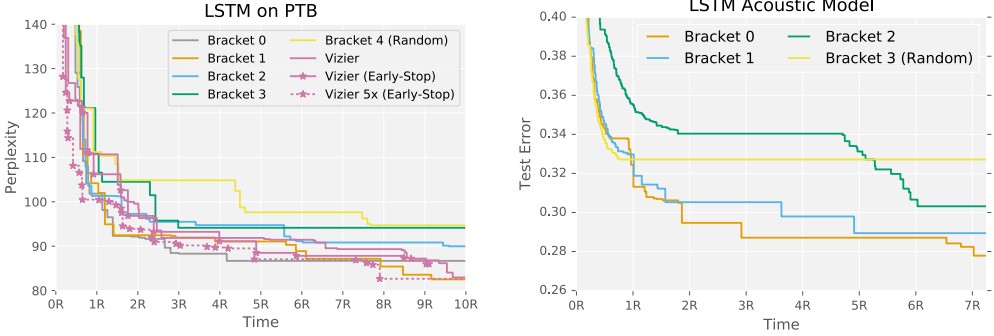

Figure 3: **Large-scale experiments** that take on the order of weeks to run. The x-axis is measured in units of average time to train a configuration, i.e. $4R$ indicates $4\times$ the time to train an average configuration. Due to the high computational cost, progress for a single trial is shown in each chart.

We conduct empirical studies on two large-scale benchmarks that take on the order of weeks to run. Both benchmarks show that asynchronous SHA is well suited for the large-scale regime and can successfully find good hyperparameter settings under resource and time constraints. We consider brackets of asynchronous SHA with $\eta = 4$ and a bounded resource per configuration $R$, with $time(R)$ representing the average time to train a single model. We compare to Vizier with and without the performance curve early-stopping rule (Golovin et al., 2017).[1] For both tasks, the resources allocated to different configurations is the number of training records, which translates into the number of

---

[1] In both cases, we use the default implementation of Vizier, which automatically selects the tuning method based on the number of function evaluations (Golovin et al., 2017).

training iterations after accounting for certain hyperparameters. Finally, due to the computational burden of these experiments, we perform only a single trial of each tuner. Our results on these tasks are qualitatively similar to the results in Section 4.2, which are averaged over multiple trials.

For the first task, we tune a one layer LSTM language model for next word prediction on the Penn Treebank (PTB) dataset (Marcus et al., 1993). The benchmark has 10 hyperparameters that control both the model architecture as well as the optimization routine (see Appendix A.2). Each tuner is given 500 workers and $10 \times time(R)$. We set the minimum resource to $r = R\eta^{-4}$, leading to 5 possible brackets of SHA. The results in Figure 3 show that the brackets with more aggressive early-stopping rates outperform random search. Bracket 0 found a good configuration for this task in $3 \times time(R)$. By that time, random search (Bracket 4) had evaluated 1.5k configurations, compared to 1.8k for Vizier, 2.3k for Vizier with early-stopping, and 52k for Bracket 0, the bracket with the most aggressive early-stopping rate. Our results show bracket 0 and bracket 1 are competitive with Vizier, despite being much simpler and easier to implement. Additionally, whereas Vizier (Early-Stop)[2] uses the heuristic performance curve early-stopping method introduced by Golovin et al. (2017), SHA offers a way to perform principled early-stopping. Additionally, for a fair comparison of parallel Hyperband to Vizier, we also show the performance of Vizier $5 \times$ (Early-Stop), which has 2.5k workers, i.e. $5 \times$ the resources as that of a single bracket. Remarkably, parallel Hyperband, via bracket 1, matches the performance of Vizier $5 \times$ (Early-Stop) without any of the optimization overhead associated with Bayesian methods, using simple random sampling and adaptive resource allocation.

Following Li et al. (2017), we evaluate the performance of SHA relative to known results for PTB using similar architectures. The architectures considered in our search space is most similar to the 2-layer LSTMs studied by Zaremba et al. (2014). Specifically, our architectures differ from those considered by Zaremba et al. (2014) only in the number of hidden LSTM nodes and the embedding dimension, both of which range from 100 to 1000 in our search space. The medium model considered by Zaremba et al. (2014) with 650 hidden LSTM nodes, which is arguably the model closest to those in our search space, reached a test perplexity of 82.7, while the best model found by bracket 1 in our experiment converged to a test perplexity of 81.3. Although the search space we used is not directly comparable to the medium model, we believe this is an encouraging result in terms of achieving state-of-the-art on this specific dataset and model family.

The acoustic modeling task trains an LSTM on a small collection of 250 thousand spoken recordings of anonymized and aggregated typed search queries. The search space we consider has 8 hyperparameters and includes models with up to 5 LSTM layers (see Appendix A.2 for details). Our model uses the LSTM cell introduced in Sak et al. (2014) with a recurrent projection layer. Each tuning method is given 20 workers and allowed to run for $7 \times time(R)$. Note that the search space includes models that take nearly the entire time allowance to train. The minimum resource for SHA is set to $r = R\eta^{-3}$ for a total of 4 possible brackets. The results in Figure 3 show a similar relative ordering of the brackets of SHA.[3] Bracket 0 found a configuration with error rate below 30% in $2 \times time(R)$ and evaluated over one thousand configurations in that time with just 20 workers.

While we have demonstrated asynchronous SHA to achieve comparable performance to Vizier, we hypothesize that even better results relative to Bayesian methods are possible in settings with more parallel resources and wall-clock times restricted to roughly the training time of a single model. Asynchronous SHA is massively scalable because it faces no degradation in performance when compensating for time with more parallel resources. In contrast, Bayesian methods are hampered in wall-clock time constrained settings, since they require updates to the posterior to effectively select new configurations.

### 4.4 PRACTICAL CONSIDERATIONS FOR HYPERPARAMETER TUNING IN PARALLEL SETTINGS

In the CNN tuning tasks from from Section 4.2, we used one GPU to train each model. However, distributed resources can also be used for parallel training to speed up the time it takes to train a single model (Krizhevsky, 2014; Szegedy et al., 2014; You et al., 2017; You et al., 2017; Goyal et al., 2017). Unfortunately, the speedups do not scale linearly with the number of GPUs due to communication costs and other overheads. Therefore, there is a tradeoff between using distributed resources to train a model faster versus to evaluate more configurations.

---

[2]Our final publication will extend Vizier with early-stopping to $10 \times time(R)$.

[3]The results for Vizier are still pending at the time of submission but will be included in the final publication.

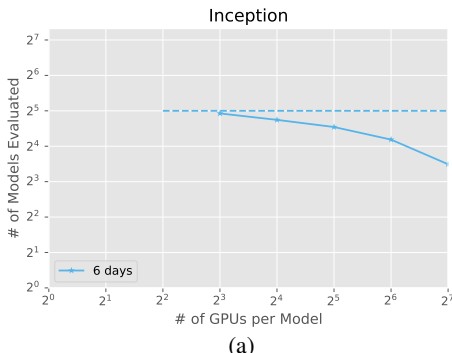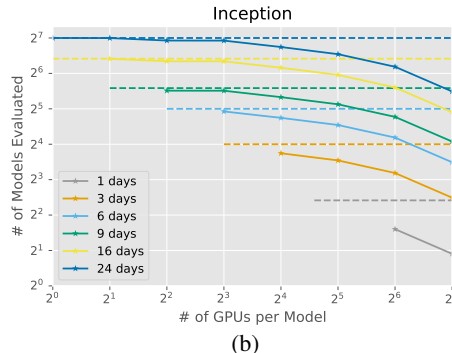

Figure 4: For a fixed time budget, we chart the estimated number (according to the Paleo performance model) of configurations evaluated by 128 Tesla K80s as a function of the number of GPUs used to train each model. The dashed line for each color represents the number of models evaluated under perfect scaling, i.e. $n$ GPUs train a single model $n$ times as fast, and span the feasible range for number of GPUs per model in order to train within the allocated time budget. (a) Imagenet using Inception V3 takes 24 days to train on a single GPU. Hence, for a time budget of 6 days, the number of GPUs per model must offer at least $4\times$ speedup over a single GPU in order to be feasible. Under perfect scaling, $128 = 2^7$ GPUs with $1/4$ the time to train a model on a single GPU would be able to train $2^5$ models. The curve shows that using $8 = 2^3$ GPUs per model achieves nearly linear scaling and increasing the number of GPUs per model decreases the total number of models that can be evaluated by 128 GPUs. (b) The same tradeoff curve for other time budgets is shown. As expected, smaller time budgets requires more GPUs per model, but the additional training speed comes at the expense of fewer evaluated models.

Qi et al. (2017) introduced Paleo, a framework for estimating the cost of training neural networks in parallel given different specifications. Qi et al. (2017) demonstrated that the speedups computed using Paleo are fairly accurate when compared to the actual observed speedups for a variety of models. Therefore, we use Paleo to estimate the training time for Inception (et al., 2016) under strong scaling (i.e. fixed batch size with increasing parallelism), Butterfly AllReduce communication scheme, standard hardware settings (namely Tesla K80 GPU and 20G Ethernet), and a batch size of 1024. We then use the estimates to produce the tradeoff curves in Figure 4. The figure shows a general trend of diminishing returns associated with using more GPUs to train a single model and suggest choosing a time budget and level of parallelism as close to to the upper left as possible (i.e. if time permits, use only one GPU per model to explore as many models as possible).

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

## A  EXPERIMENT DETAILS

In the below sections, we detail the experimental setup for the benchmarks in Section 4. The SVM task uses the same search space and experimental setup as Klein et al. (2017a) and the cuda-convnet task is the same as that in Li et al. (2017).

### A.1  EXPERIMENTAL SETUP FOR THE SMALL CNN ARCHITECTURE TUNING TASK

| Hyperparameter | Type | Values |
|---|---|---|
| batch size | choice | $\{2^6, 2^7, 2^8, 2^9\}$ |
| # of layers | choice | $\{2, 3, 4\}$ |
| # of filters | choice | $\{16, 32, 48, 64\}$ |
| weight init std 1 | continuous | $\log [10^{-4}, 10^{-1}]$ |
| weight init std 2 | continuous | $\log [10^{-3}, 1]$ |
| weight init std 3 | continuous | $\log [10^{-3}, 1]$ |
| $l_2$ penalty 1 | continuous | $\log [10^{-5}, 1]$ |
| $l_2$ penalty 2 | continuous | $\log [10^{-5}, 1]$ |
| $l_2$ penalty 3 | continuous | $\log [10^{-3}, 10^2]$ |
| learning rate | continuous | $\log [10^{-5}, 10^1]$ |

Table 1: Hyperparameters for small CNN architecture tuning task.

This benchmark tunes a multiple layer CNN network with the hyperparameters shown in Table 1. The # of layers hyperparameter indicate the number of convolutional layers before two fully connected layers. The # of filters indicates the # of filters in the CNN layers with the last CNN layer having $2 \times \#$ filters. Weights are initialized randomly from a Gaussian distribution with the indicated standard deviation. There are three sets of weight init and $l_2$ penalty hyperparameters; weight init 1 and $l_2$ penalty 1 apply to the convolutional layers, weight init 2 and $l_2$ penalty 2 to the first fully connected layer, and weight init 3 and $l_2$ penalty 3 to the last fully connected layer. Finally, the learning rate hyperparameter controls the initial learning rate for SGD. All models use a fixed learning rate schedule with the learning rate decreasing by a factor of 10 twice in equally spaced intervals over the training window. This benchmark is run on the SVHN dataset (Netzer et al., 2011) following Sermanet et al. (2012) to create the train, validation, and test splits.

### A.2  EXPERIMENTAL SETUP FOR LARGE-SCALE BENCHMARKS

The LSTM models tuned for our two large-scale benchmarks are implemented in Tensorflow. There are a few sources of randomness that impact the relative performance of the tuning methods:

1. Randomly sampled hyperparameter configurations.
2. Random weight initialization.
3. Random minibatches.

We control for these sources to the best of our ability but randomness remain in part due to multi-threaded data input queues and non-associative floating point operations.

The hyperparameters for the LSTM tuning task on the Penn Tree Bank (PTB) dataset presented in Section 4.3 is shown in Table 2. Note that all hyperparameters are tuned on a linear scale and sampled uniform over the specified range. The inputs to the LSTM layer are learned embeddings of the words in a sequence. The number of hidden nodes hyperparameter refers to the number of nodes in the LSTM. The learning rate is decayed by the decay rate after each interval of decay steps. Finally, the weight initialization range indicates the upper bound of the uniform distribution used to initialize all weights. The other hyperparameters have their standard interpretations for neural networks. The default training (929k words) and test (82k words) splits for PTB are used for training and evaluation (Marcus et al., 1993).

The hyperparameters for the acoustic model tuning task presented in Section 4.3 is shown in Table 3. The hyperparameters that overlap with the PTB task also have the same interpretation. The only additional hyperparameter is the projection fraction, which indicates the size of the recurrent projection

| Hyperparameter | Type | Values |
|---|---|---|
| batch size | choice | $\{5, 10, 20, 30, 40, 50, 60, 80\}$ |
| embedding dims | discrete | $[100, 1000]$ |
| # of time steps | discrete | $[10, 100]$ |
| # of hidden nodes | discrete | $[100, 1000]$ |
| learning rate | continuous | $[1, 100]$ |
| decay rate | continuous | $[0.01, 0.99]$ |
| decay steps | discrete | $[1000, 10000]$ |
| clip gradients | continuous | $[1, 5]$ |
| dropout probability | continuous | $[0.1, 0.9]$ |
| weight init range | continuous | $[0.1, 0.9]$ |

Table 2: Hyperparameters for PTB LSTM task.

layer in the LSTM cell introduced by Sak et al. (2014) as a fraction of the number of hidden units in the LSTM. The model is trained on a dataset with 250k examples and evaluated on a test set with 2k examples.

| Hyperparameter | Type | Values |
|---|---|---|
| batch size | choice | $\{1, 2, 4, 8, 12, 18, 24, 30, 40, 50, 60\}$ |
| # of time steps | choice | $\{10, 20, 50\}$ |
| # of LSTM layers | choice | $\{1, 3, 5\}$ |
| # of hidden nodes | choice | $\{8, 32, 64, 128, 256, 512, 1024\}$ |
| projection fraction | choice | $\{.125, .25, .375, .5\}$ |
| learning rate | continuous | $\log [5e{-}10, 5e{-}1]$ |
| decay rate | continuous | $[0.01, 0.99]$ |
| decay steps | discrete | $\log [10000, 120000000]$ |

Table 3: Hyperparameters for acoustic model LSTM task.

