# OpenReview forum: "Massively Parallel Hyperparameter Tuning"
_ICLR.cc/2018/Conference — Reject_

### Official Review · AnonReviewer3 · 2017-11-27
**A good extension of Hyperband to allow for parallel evaluation, but I have a few questions to clear up.**

**Rating:** 5
**Confidence:** 5

**Review:**

In this paper, the authors extend Hyperband--a recently proposed non model based hyperparameter tuning procedure--to better support parallel evaluation. Briefly, Hyperband builds on a "successive halving" algorithm. This algorithm allocates a budget of B total time to N configurations, trains for as long as possible until the budget is reached, and then recurses on the best N/2 configurations--called the next "rung" in the paper. Thus, as optimization proceeds, more promising configurations are allowed more time to train. This basic algorithm has the problem that different optimization tasks may require different amounts of time to become distinguishable; Hyperband solves this by running multiple rounds of succesive halving--called "brackets"--varying the initial conditions. That is, should successive halving start with more initial configurations (but therefore less budget for each configuration), or a small number of configurations. The authors further extend Hyperband by allowing the successive halving algorithm to be run in parallel. To accomplish this, when a worker looks for a job it prefers to run jobs on the next available rung; if none are currently outstanding, a new job is started on the lowest rung.

Overall, I think this is a natural scheme for parallelzing Hyperband. It is extremely simple (a good thing), and neatly circumvents the obvious problem with parallelizing Hyperband, which is that successive halving naturally limits the number of jobs that can be done. I think the non-model based approach to hyperparameter tuning is compelling and is of interest to the AutoML community, as it raises an obvious question of how approaches that exploit the fact that training can be stopped any time (like Hyperband) can be combined with model-based optimization that attempt to avoid evaluating configurations that are likely to be bad.

However, I do have a few comments and concerns for the for the authors to address that I detail below. I will be more than happy to modify my evaluation if these concerns are addressed by the authors.

First and most importantly, can the authors discuss the final results achieved by their hyperparameter optimization compared to state-of-the-art results in the field? I am not sure what SOTA is on the Penn Treebank  or acoustic modeling task, but obviously the small ConvNet getting 20% error on CIFAR10 is not state of the art. Do the authors think that their technique could improve SOTA on CIFAR10 or CIFAR100 if applied to a modern CNN architecture like a ResNet or DenseNet?

Obviously these models take a bit longer to train, but with the ability to train a large number of models in parallel, a week or two should be sufficient to finish a nontrivial number of iterations. The concern that I have is that we continue to see these hyperparameter tuning papers that discuss how important the task is, but--to the best of my knowledge--the last paper to actually improve SOTA using automated hyperparameter tuning was Snoek et al., 2012., and there they even achieved 9.5% error with data augmentation. Are hyperparameters just too well tuned on these tasks by humans, and the idea is that Hyperband will be better on new tasks where humans haven't been working on them for years? In BayesOpt papers, hyperparameter tuning has often been used simply as a task to compare optimization performance, but I don't think this argument applies to Hyperband because it isn't really applicable to blackbox functions outside of hyperparameter tuning because it explicitly relies on the fact that training can be cut short at any time.

Second (and this is more of a minor point), I am a little baffled by Figure 4. Not by the argument you are trying to make--it of course makes sense to me that additional GPUs would result in diminishing returns as you become unable to fully exploit the parallelism--but rather the plots themselves. To explain my confusion, consider the 8 days curve in the AlexNet figure. I read this as saying, with 1 GPU per model, in 8 days, I can consider 128 models (the first asterisk). With 2 GPUs per model, in 8 days, I can consider slightly less than 128 models (the second asterisk). By the time I am using 8 GPUs per model, in 8 days, I can only train a bit under 64 models (the fourth asterisk). The fact that these curves are monotonically decreasing suggests that I am just reading the plot wrong somehow -- surely going from 1 GPU per model to 2 should improve performance somewhere? Additionally, shouldn't the dashed lines be increasing, not horizontal (i.e., assuming perfect parallelism, if you increase the number of GPUs per model--the x axis--the number of models I can train in 8 days--the y axis--increases)?

---

### Official Review · AnonReviewer1 · 2017-11-27
**Small speedup by parallelization?**

**Rating:** 5
**Confidence:** 5

**Review:**

This paper introduces a simple extension to parallelize Hyperband.

Points in favor of the paper:
* Addresses an important problem

Points against:
* Only 5-fold speedup by parallelization with 5 x 25 workers, and worse performance in the same budget than Google Vizier (even though that treats the problem as a black box)
* Limited methodological contribution/novelty


The paper's methodological contribution is quite limited: it amounts to a straight-forward parallelization of successive halving (SHA). Specifically, whenever a worker frees up, do a new run on it, at the highest rung possible while making sure to not run too many runs for too high rungs. (I am pretty sure that is the idea, even though Algorithm 1, which is supposed to give the details, appears to have a bug in Procedure get_job -- it would always either pick the highest rung or the lowest!)

Empirically, the paper strangely does not actually evaluate a parallel version of Hyperband, but only evaluates the 5 parallel variants of SHA that Hyperband would run, each of them with all workers. The experiments in Section 4.2 show that, using 25 workers, the best of these 5 variants obtains a 5-fold speedup over sequential Hyperband on CIFAR and an 8-fold speedup on SVHN. I am confused: the *best* of 5 SHA variants only achieves a 5-fold speedup using 25 workers? I.e., parallel Hyperband, which would run the 5 SHA variants in parallel, would require 125 workers but only yield a 5-fold speedup? If I understand this correctly, I would clearly call this a negative result.

Likewise, for the large-scale experiment, a single run of Vizier actually yields as good performance as the best of the 5 SHA variants, and it is unknown beforehand which SHA variant works best -- in this example, actually Bracket 0 (which is often the best) stagnates. Parallel Hyperband would run the 5 SHA variants in parallel, so its performance at a budget of 10R with a total of 500 workers can be evaluated by taking the minimum of the 5 SHA variants at a budget of 2R. This would obtain a perplexity of above 90, which is quite a bit worse than Vizier's result of about 82. In general, the performance of parallel Hyperband can be computed by taking the minimum of the SHA variants and multiplying the time taken by 5; this shows that at any time in the plot (Figure 3, left) Vizier dominates parallel Hyperband. Again, this is apparently a negative result. (For Figure 3, right, no results for Vizier are given yet.)

If I understand correctly, the experiment in Section 4.4 does not involve any run of Hyperband, but merely plots predictions of Qi et al.'s Paelo framework of how many models could be evaluated with a growing number of GPUs.

Therefore, all empirical results for parallel Hyperband reported in the paper appear to be negative. This confuses me, especially since the authors seem to take them as positive results.
Because the original Hyperband paper argued that Bayesian optimization does not parallelize as well as random search / Hyperband, and because Hyperband has been reported to work much better than Bayesian optimization on a single node, I would have expected clear improvements of parallel Hyperband over parallel Bayesian optimization (=Vizier in the authors' setup). However, this is not what I see in the results. Am I mistaken somewhere? If not, based on these negative results the paper does not seem to quite clear the bar for ICLR.


Details, in order of appearance in the paper:

- Vizier: why did the authors only use Vizier's default Bayesian optimization algorithm? The Vizier paper by Golovin et al (2017) states that for large budgets other optimizers often perform better, and the budget in the large scale experiments is as high as 5000 function evaluations. Also, isn't there an automatic choice built into Vizier to pick the optimizer expected to be best? I think using a suboptimal version of Vizier would be a problem for the experimental setup.
- Algorithm 1: this needs some improvement; in particular fixing the bug I mentioned above.
- Section 3.1: Li et al (2017) do not analyze any algorithm theoretically. They also do not discuss finite vs. infinite horizon. I believe the authors meant Li et al's arXiv paper (2016) in both of these cases.
- Section 3.1, point 2: this is unclear to me, even though I know Hyperband very well. Can you please make this clearer?
- "A complete theoretical treatment of asynchronous SHA is out of the scope of this paper" -> is some theoretical treatment in scope?
- Section 4.1: It seems very useful to already recommend configurations in each rung of Hyperband, and I am surprised that the methods section does not mention this. From the text in this experiments section, it feels a little like that was always part of Hyperband; I didn't think it was, so I checked the original papers and blog posts, and both the ICLR 2017 and the arXiv 2016 paper state "In fact, the first result returned by HYPERBAND after using a budget of 5R is often competitive with results returned by other searchers after using 50R." and Kevin Jamieson's blog post on Hyperband (https://people.eecs.berkeley.edu/~kjamieson/hyperband.html) explicitly states: "While random and the Bayesian Optimization algorithms output their first recommendation after max_iter iterations, Hyperband does not output anything until about max_iter(logeta(max_iter)+1) iterations [...]"
Therefore, recommending after each rung seems to be a contribution of this paper, and I think it would be nice to read about this in the methods section.
- Experiment 1 (SVM) used dataset size as a budget, which is what Fabolas ("Fast Bayesian optimization on large datasets") is designed for according to Klein et al (2017). On the other hand, Experiments (2) and (3) used the number of epochs as a budget, and Fabolas is not designed for that (one would want to use a different kernel, for epochs, e.g., like Freeze-Thaw Bayesian optimization (FTBO) by Swersky et al (2014), instead of a kernel made for dataset sizes). Therefore, it is not surprising that Fabolas does not work as well in those cases. The case of number of epochs as a budget would be the domain of FTBO. I know that there is no reference implementation of FTBO, so I am not asking for a comparison, but the comparison against Fabolas is misleading for Experiments (2) and (3). This doesn't really change anything for the paper: the authors could still make the case that Fabolas hasn't been designed for this case and that (to the best of my knowledge) there simply isn't an implementation of a BO algorithm that is. Fabolas is arguably the closest thing, so the results could still be reported, just not as an apples-to-apples comparison; probably best as "Fabolas-like, with dataset size kernel" in the figure. The justification to not compare against Fabolas in the parallel regime is clearly valid.
- A clarification question: Section 4.4 does not report on any runs of actual neural networks, does it? And not on any runs of Hyperband, correct? Do I understand the reasoning correctly as pointing out that standard parallelization across multiple GPUs is not great, and that thus, in combination with parallel Hyperband, runs should be done mostly on one GPU only? How does this relate to the results in the cited paper "Accurate, Large-batch SGD: Training ImageNet in 1 Hour" (https://arxiv.org/abs/1706.02677)? Quoting from its abstract: "Using commodity hardware, our implementation achieves ∼ 90% scaling efficiency when moving from 8 to 256 GPUs." That seems like a very good utilization of parallel computing power?
- There is no conclusion / future work.

----------
Edit after author rebuttal:
I thank the reviewers for their rebuttal. This cleared up some points, but some others are still open.
(1) and (2) Unfortunately, I still do not agree that the need for 5*25 workers to get a 5-fold to 8-fold speedup is a positive result. Similarly, I would interpret the results in Figure 3 differently than the authors. For the comparison against Vizier the authors argue that they could just take the lowest 2 brackets of Hyperband; but running both of these two would still be 2x slower than Vizier. And we can't only run the best bracket because the information which one is the best is not available ahead of time. In fact, it is the entire point of Hyperband to hedge across multiple brackets including the one that is random search; one *could* just use the smallest bracket, but that is a heuristic and has no theoretical guarantees of being better (or at least not worse by more than a bounded factor) than random search.
Orthogonally: the comparison to Vizier (or any other baseline) is still missing for the LSTM acoustic model.

(3) Concerning SOTA results, I have to agree with AnonReviewer3: one way to demonstrate success is to show competitive performance on a dataset (e.g., CIFAR) on which other researchers can also evaluate their algorithms on. Getting 17% on CIFAR-10 does not fall into that category. Nevertheless, I agree with the authors that another way to demonstrate success is to show competitive performance on a *combination* of a dataset and a design space, but for that to be something that other researchers can compare to requires the authors making publicly available the implementations they have optimized; without that public availability, due to a host of possible confounding factors, it is impossible to judge whether state-of-the-art performance on such a combination of dataset and design space has been achieved.  I therefore recommend that the authors make the entire code they used for training CIFAR available; I don't expect this to have anything new in there, but it's a useful benchmark.
Likewise, for the LSTM on PTB, DeepMind used Google Vizier (https://arxiv.org/abs/1707.05589) to achieve *perplexities below 60* (compared to the results above 80 reported by the authors). Just as above, I therefore recommend that the authors make their pipeline for LSTB on PTB available. Likewise for the LSTM acoustic model.

(4) I'm confused that Section 4.4 does relate to SHA/Hyperband. Of course, there are some diminishing returns of running an optimizer across multiple GPUs. But similarly, there are diminishing returns of parallelizing SHA (e.g., the 5-fold speedup on 125 workers above). So the natural question that would be nice to answer is which combination of the two will yield the best results. Relatedly, the paper by Goyal et al seems to show that the weak scaling regime leads to almost linear speedups; why do the authors then analyze the strong scaling regime that does not appear to work as well?

Overall, the rebuttal did not change my evaluation and I kept my original score.

---

### Official Review · AnonReviewer2 · 2017-11-29
**Adapting hyperband to run on a cluster**

**Rating:** 6
**Confidence:** 3

**Review:**

This paper adapts the sequential halving algorithm that underpins Hyperband to run across multiple workers in a compute cluster. This represents a very practical scenario where a user of this algorithm would like to trade off computational efficiency for a reduction in wall time. The paper's empirical results confirm that indeed significant reductions in wall time come with modest increases in overall computation, it's a practical improvement.

The paper is crisply written, the extension is a natural one, the experiment protocols and choice of baselines are appropriate.

The left panel of figure 3 is blurry, compared with the right one.

---

### Author Response · Authors · 2017-12-30
**Response to Comments from Reviewers**

We thank the reviewers for providing thoughtful comments and feedback for our paper.  In this rebuttal, we focus on 4 main topics:

(1) Wall-Clock Problem: As noted in the introduction, we are tackling the problem of evaluating “orders of magnitude more hyperparameter configurations than available parallel workers in a small multiple of the wall-clock time needed to train a single model.”  As a result, in our experiments, we focus on the raw magnitude of the time taken to return an accurate model, rather than speedups versus the sequential setting.

(2) Successive Halving versus Hyperband: For various reasons described below, we believe that comparing Vizier with Successive Halving is justified. That said, we also see the merit in comparing against Hyperband, and we have just completed running an additional experiment to this end.

(3) State-of-the-art (SOTA) Results: Similar to the original Hyperband work by Li, et. al. (2017), we evaluate the relative performance of asynchronous SHA by comparing to published results on the same dataset (e.g., CIFAR10) *and* the same model family (e.g., cudaconvnet with a specific search space). Under this evaluation criterion, our proposed methods are SOTA on CIFAR10/cudaconvnet and are quite promising on PTB for our specified search space.

(4) Training with Multiple GPUs (Section 4.4 / Figure 4): The goal of this section is to explore how to trade-off between distributed training and embarrassingly parallel hyperparameter optimization when working on GPU clusters. While our underlying arguments have not changed, we will update the text and the figure to clarify our arguments / improve the presentation.

***Note:*** We are still working on the final touches of our updated draft, and will upload the new PDF in the next few days.

---

> ### Author Response · Authors · 2017-12-30
> **Other Comments**
>
> - (Reviewer 1) “Limited methodological contribution/novelty”
>
> We respectfully disagree with the reviewer’s comment.  While our resulting algorithm is simple, we certainly did not start out with this approach.  We initially tried to parallelize Successive Halving by parallelizing each bracket rung by rung.  However, this approach suffers from a geometrically decreasing number of jobs as the algorithm progresses to higher rungs, and is highly susceptible to stragglers and dropped jobs.  While the first issue can be addressed by starting more brackets as the number of jobs decrease, the second issue is more serious and harder to address.  We also considered taking a divide and conquer approach, i.e. running a separate bracket of Successive Halving on each machine.  However, this approach is poorly suited for the wallclock-time constrained setting, because it takes the same amount of time as the sequential algorithm to return a configuration trained to completion.  In contrast, the algorithm we present addresses all of these issues, while retaining the benefits of Successive Halving.  Moreover, as noted by Reviewer 3, we view the simplicity of our resulting algorithm as one of its core benefits.
>
> - (Reviewer 1) “Section 3.1, point 2: this is unclear to me, even though I know Hyperband very well. Can you please make this clearer?”
>
> The algorithm analyzed by Li, et. al. (2016) doubles the budget for SHA in the outer loop, thereby increasing the starting number of configurations for each bracket of SHA.  This is required by the theoretical analysis, which assumes that eventually the optimal bracket of SHA will be run with enough configurations so that an epsilon good configuration is in the starting set with high probability.  In contrast, the actual algorithm that Li, et. al. (2016) recommend using in practice keeps the starting number of configurations constant to prevent doubling waiting times in between configurations trained to completion.  Our algorithm addresses this discrepancy by  naturally grows the number of configurations without blocking promotions to the top rung.
>
> - (Reviewer 1) “It seems very useful to already recommend configurations in each rung of Hyperband, and I am surprised that the methods section does not mention this.” ... “Therefore, recommending after each rung seems to be a contribution of this paper, and I think it would be nice to read about this in the methods section.”
>
> We agree that intermediate losses observed by SHA in the lower rungs are already very useful for selecting a good configuration. While straightforward, this is not mentioned in the original Hyperband paper, and we will make note of this in our updated submission.
>
> - (Reviewer 1) Bug in Algorithm 1.
>
> We apologize for the confusing wording used in Algorithm 1.  The algorithm simply moves from the top rung to lower rungs in search of a promotable configuration.  We will rephrase the wording in Algorithm 1 to reflect this.

---

> ### Author Response · Authors · 2017-12-30
> **(3) SOTA Results and (4) Training with Multiple GPUs**
>
> (3) State-of-the-art Results:
>
> - (Reviewer 3) “[C]an the authors discuss the final results achieved by their hyperparameter optimization compared to state-of-the-art results in the field?” ... “Do the authors think that their technique could improve SOTA on CIFAR10 or CIFAR100 if applied to a modern CNN architecture like a ResNet or DenseNet?” ... “Are hyperparameters just too well tuned on these tasks by humans, and the idea is that Hyperband will be better on new tasks where humans haven't been working on them for years?”
>
> Li, et. al. (2017) compared the performance of Hyperband to state-of-the-art hyperparameter tuning results on CIFAR-10 using the cudaconvnet architecture.  When limited to this architecture, Hyperband was able to achieve state-of-the-art (SOTA) results, exceeding the accuracy of the hand-tuned model by ~1% point.  We hypothesize that SHA/Hyperband can improve upon SOTA on more modern CNN architectures as well, since it would be able to explore the space much faster than popular methods like random search and hand-tuning.
>
> For the PTB experiment in Section 4.3, our model family and search space encompasses the 2-layer LSTMs studied by Zaremba, et. al. (2015).  Specifically, our range for # of hidden LSTM nodes of 10 to 1000 is closest to the medium model with 650 nodes considered by Zaremba, et. al. (2015).  Their medium model achieved a test perplexity of 82.7 while the best model found by bracket 1 in our experiment achieved a test perplexity of 81.3.  Although the search space we used is not directly comparable to the medium model, we believe this is an encouraging result in terms of achieving SOTA on this specific dataset and model family.  We will add a paragraph to the revision discussing the performance of our results relative to other published works.
>
> - (Reviewer 1) “Vizier: why did the authors only use Vizier's default Bayesian optimization algorithm?”
>
> The default Vizier algorithm that we used automatically selects the Bayesian optimization method that is expected to perform the best.  Batched Gaussian Process Bandits is used when the number of trials is under 1k, otherwise Vizier uses a proprietary local-search algorithm (Golovin, et. al., 2017).
>
> (4) Training with Multiple GPUs (Section 4.4 / Figure 4):
>
> - (Reviewer 1) “A clarification question: Section 4.4 does not report on any runs of actual neural networks, does it? And not on any runs of Hyperband, correct?”
>
> Yes, this is correct.  All numbers reported in Section 4.4. are *predicted* results using the PALEO performance model.
>
> - (Reviewer 1) “Do I understand the reasoning correctly as pointing out that standard parallelization across multiple GPUs is not great, and that thus, in combination with parallel Hyperband, runs should be done mostly on one GPU only? How does this relate to the results in the cited paper ‘Accurate, Large-batch SGD: Training ImageNet in 1 Hour" (https://arxiv.org/abs/1706.02677)? Quoting from its abstract: "Using commodity hardware, our implementation achieves ∼ 90% scaling efficiency when moving from 8 to 256 GPUs.’ That seems like a very good utilization of parallel computing power?”
>
> We measure speedups in terms of time per iteration under a *strong scaling regime*, where batch size stays the same as the # of GPUs used to train a single model increases.  This results in a serial-equivalent execution.  In contrast, the paper (Goyal, et al.) referenced by the reviewer focuses on the weak scaling regime, where batch size increases with # of GPUs per model.  Weak scaling is of course advantageous from a computational perspective, however, it does not preserve serial-equivalency, and the resulting models can often underperform (in terms of accuracy) relative to models trained using smaller batches.  Goyal, et. al. was impressively able to achieve high accuracy on ImageNet with Resnet50 while using a  large batch sizes by adjusting the learning rate, which certainly makes the use of large batch sizes/weak scaling for distributed training more appealing. However, it is unclear whether these techniques generalize to other tasks and models. Furthermore, the result in Goyal et. al does not invalidate either the specific results we present or the more general point about the attractiveness of leveraging distributed resources in an embarrassingly parallel fashion in the context of problems requiring hyperparameter optimization.
>
>
> - (Reviewer 3) Confusion regarding Figure 4.
>
> We will improve the chart as well as the caption in Figure 4 to simplify the presentation. The lines show how many different configurations/models can be evaluated for a fixed number of GPUs (128 = 2^7 Tesla K80s) and a given time budget.  A higher number of GPUs per model is required when we want to train an individual model faster. However, since training speed does not scale linearly with # of GPUs per model, training a model faster comes at the expense of fewer total models trained.

---

> ### Author Response · Authors · 2017-12-30
> **(1) Wall-Clock Problem and (2) Successive Halving vs Hyperband**
>
> (1) Wall-Clock Problem:
>
> - (Reviewer 1) “The experiments in Section 4.2 show that, using 25 workers, the best of these 5 variants obtains a 5-fold speedup over sequential Hyperband on CIFAR and an 8-fold speedup on SVHN. I am confused: the *best* of 5 SHA variants only achieves a 5-fold speedup using 25 workers? I.e., parallel Hyperband, which would run the 5 SHA variants in parallel, would require 125 workers but only yield a 5-fold speedup? If I understand this correctly, I would clearly call this a negative result.”
>
> The small scale experiment in Section 4.2 demonstrates that our algorithm succeeds in the wall-clock constrained setting, which is precisely the focus of this paper.  We show that asynchronous SHA finds a good configuration for this benchmark in the time to train one model to completion.  For these experiments, we observed only 5-8x speedup for 25 workers because a few hundred configurations were sufficient to find a good setting and exploring over 6k configurations with 25 workers did not offer significant benefit.  Hence, most of the speedup can be attributed to reducing the time for training a single configuration to completion from 5R in the sequential setting to ~1R for asynchronous SHA (which again is the stated goal of this work).
>
> (2) Successive Halving versus Hyperband:
>
> - (Reviewer 1) “Likewise, for the large-scale experiment, a single run of Vizier actually yields as good performance as the best of the 5 SHA variants, and it is unknown beforehand which SHA variant works best -- in this example, actually Bracket 0 (which is often the best) stagnates. Parallel Hyperband would run the 5 SHA variants in parallel, so its performance at a budget of 10R with a total of 500 workers can be evaluated by taking the minimum of the 5 SHA variants at a budget of 2R. This would obtain a perplexity of above 90, which is quite a bit worse than Vizier's result of about 82. In general, the performance of parallel Hyperband can be computed by taking the minimum of the SHA variants and multiplying the time taken by 5; this shows that at any time in the plot (Figure 3, left) Vizier dominates parallel Hyperband. Again, this is apparently a negative result.”
>
> In this work, we focused on generalizing Successive Halving (SHA) for the parallel setting.  As stated in the end of Section 3, we believe that in practice, users often try one or two brackets of Successive Halving that perform aggressive early-stopping.  This is supported by Li, et. al. (2017), which showed that brackets 0 and 1 were the top two performing brackets in all their experiments covering a wide array of hyperparameter tuning tasks.  We also found this to be true in our experiments, and in our experience, aggressive early-stopping is highly effective, especially for neural network tasks. Moreover, for modern day hyperparameter tuning problems with high-dimensional search spaces and models that are very expensive to train, aggressive early-stopping is necessary for the problem to be tractable.
>
> In light of these arguments, we believe it is reasonable to compare Vizier to just the most aggressive brackets of SHA.  Our results show bracket 0 and bracket 1 are competitive with Vizier, despite being much simpler and easier to implement.  Additionally, whereas the nonparametric regression early-stopping method used in Vizier (based on the work in Golovin, et. al. (2017)) is heuristic in nature, SHA offers a way to perform principled early-stopping.
>
> That said, we agree with the reviewer that a comparison to Hyperband would be informative. Hence, we have performed additional experiments for “Vizier 5x (Early-Stop)”, which represents Vizier with early-stopping run with 5 times the resources (i.e. 2.5k workers, which is the same as that used for 5 brackets of SHA).  We will add this to the chart with the PTB results in Figure 3.  The results show that Vizier 5x (Early-Stop) performs comparably to brackets 0 and bracket 1, and hence to parallel Hyperband, which takes the minimum across all 5 brackets.  Remarkably, Hyperband matches the performance of Vizier 5x without any of the optimization overhead associated with Bayesian methods, using simple random sampling and adaptive resource allocation.
>
> - (Reviewer 1) “worse performance in the same budget than Google Vizier (even though that treats the problem as a black box)”
>
> As stated in our response to the previous comment, asynchronous SHA performs comparably to Vizier with and without early-stopping, and similarly, parallel Hyperband performs comparably to Vizier 5x with early-stopping.  We note the early-stopping variant of Vizier does not treat the problem as “black-box.”

---

### Author Response · Authors · 2018-01-02
**Revision Posted**

A revised version of our paper, incorporating the feedback from reviewers, has been posted.

---

### Decision · Program_Chairs · 2018-01-29
**ICLR 2018 Conference Acceptance Decision**

**Decision:**

Reject

**Comment:**

This paper presents a simple tweak to hyperband to allow it to be run asynchonously on a large cluster, and contains reasonably large-scale experiments.

The paper is written clearly enough, and will be of interest to anyone running large-scale ML experiments.  However, it falls below the bar by:
1) Not exploring the space of related ideas more.
2) Not providing novel insights.
3) Not attempting to compare against model-based parallel approaches.